# Hepatitis C Virus Prevalence and Risk Factors in a Village in Northeastern Romania—A Population-Based Screening—The First Step to Viral Micro-Elimination

**DOI:** 10.3390/healthcare9060651

**Published:** 2021-05-31

**Authors:** Laura Huiban, Carol Stanciu, Cristina Maria Muzica, Tudor Cuciureanu, Stefan Chiriac, Sebastian Zenovia, Vladut Mirel Burduloi, Oana Petrea, Ana Maria Sîngeap, Irina Gîrleanu, Cătălin Sfarti, Camelia Cojocariu, Anca Trifan

**Affiliations:** 1Department of Gastroenterology “Grigore T. Popa”, University of Medicine and Pharmacy, 700115 Iasi, Romania; huiban.laura@yahoo.com (L.H.); stanciucarol@yahoo.com (C.S.); lungu.christina@yahoo.com (C.M.M.); drcuciureanutudor@gmail.com (T.C.); stefannchiriac@yahoo.com (S.C.); sebastianzenovia20@gmail.com (S.Z.); stoica_oanacristina@yahoo.com (O.P.); anamaria.singeap@yahoo.com (A.M.S.); gilda_iri25@yahoo.com (I.G.); cvsfarti@gmail.com (C.S.); cameliacojocariu@yahoo.com (C.C.); 2Institute of Gastroenterology and Hepatology, “St. Spiridon” Emergency Hospital, 700111 Iasi, Romania; 3Department of Anatomy “Grigore T. Popa”, University of Medicine and Pharmacy, 700115 Iasi, Romania; vburduloi@yahoo.com

**Keywords:** micro-elimination, hepatitis C virus treatment, screening, cascade of care, health policy

## Abstract

(1) Background: The World Health Organization adopted a strategy for the Global Health Sector on Viral Hepatitis in 2016, with the main objective of eliminating hepatitis C virus (HCV) by 2030. In this work, we aimed to evaluate the prevalence of HCV infection and risk factors in a Romanian village using population-based screening as part of the global C virus eradication program. (2) Methods: We conducted a prospective study from March 2019 to February 2020, based on a strategy as part of a project designed to educate, screen, treat and eliminate HCV infection in all adults in a village located in Northeastern Romania. (3) Results: In total, 3507 subjects were invited to be screened by rapid diagnostic orientation tests (RDOT). Overall, 2945 (84%) subjects were tested, out of whom 78 (2.64%) were found to have positive HCV antibodies and were scheduled for further evaluation in a tertiary center of gastroenterology/hepatology in order to be linked to care. In total, 66 (85%) subjects presented for evaluation and 55 (83%) had detectable HCV RNA. Of these, 54 (98%) completed antiviral treatment and 53 (99%) obtained a sustained virological response. (4) Conclusions: The elimination of hepatitis C worldwide has a higher chance of success if micro-elimination strategies based on mass screening are adopted.

## 1. Introduction

Hepatitis C virus (HCV) infection represents one of the most common causes of end-stage liver disease and hepatocellular carcinoma (HCC) and represents a major public health issue, with a global death rate exceeding that of HIV, tuberculosis or malaria [1]. The global prevalence of viremic HCV was estimated to be 1% in 2015, corresponding to 71 million people worldwide, with 1.75 million new infections a year, which are associated with healthcare, drug use and blood transfusion before the screening of donors [2,3]. In Central and Eastern Europe, the prevalence of anti-HCV antibodies varies between 0.27 and 3.5%, with the number of people infected with HCV in the general population being about 1.16 million [4,5]. Romania is considered to be the country with the highest prevalence of HCV in Europe, with various reported figures in recent decades (5.9% in 1990, 3.23% in 2010 and recently estimated at 2.7%, corresponding to 550,000 patients with viral loads) [6,7]. Moldavia is the Romanian region with the highest prevalence of HCV infection [7]. An estimation of the number of people infected with HCV in the population is very important for a country’s health policy as it allows the planning of preventive and therapeutic interventions, as well as the determination of the need for the treatment of infected persons.

The introduction of direct-acting antivirals (DAAs) for the treatment of HCV infection in 2013 changed the landscape of HCV therapy from a disease that could only be cured in less than half of patients, with a long therapy process with many side effects, to a disease that can be eradicated in 8–12 weeks in almost all patients, with all-oral interferon free regimens based on DAAs [8,9]. Achieving a sustained virological response (SVR) rate of ≥95% in patients treated with DAAs has revolutionized the approach to all patients with HCV infection [9,10]. 

The costs of HCV for the global health system are significant and include the ongoing treatment of cirrhosis and HCC, amounting to an enormous expenditure by all countries [11]. In 2010, a survey that included five European countries showed that the estimated direct and indirect costs of patients with chronic hepatitis C were 76% and 65%, respectively, higher than for people without hepatitis C [12]. Moreover, HCV is accompanied by a loss of productivity among those infected, with patients proving unable to participate fully in their workplaces and in society [13].

The efficacy of the new DAA therapy has aroused immense international interest in the global elimination of HCV as a public health threat. In 2016, the World Health Organization (WHO) adopted its first-ever Global Health Sector Strategy on Viral Hepatitis—an ambitious and time-bound strategy for eliminating HCV by 2030 (a 90% reduction of new viral hepatitis infections, treatment of 80% of eligible patients and 65% reduction in liver-related deaths) [14]. To achieve this goal, the following actions should be considered: (I) ensuring access to hepatitis C treatment through the availability of treatment in smaller cities and (II) improving screening with rapid diagnostic testing (RDOT) that combines HIV, HCV and HBV [15].

To fulfil the WHO targets, all countries should have individual plans, as there is no “one size-fits-all” approach [16]. Thus, several strategies have been proposed: one is micro-elimination, a concept that breaks down the national elimination goals into smaller, easier to accomplish tasks focused on specific subpopulations (people with hemophilia or HIV, hemodialysis, transplant recipients, prisoners, people who inject drugs, men who have sex with men, migrants from high endemic countries and health care workers) as well as geographical areas (regions, cities, villages) [17]. Thus, smaller-scale policy initiatives targeting certain populations or localities are a concrete step towards the global elimination of HCV. 

We aimed to assess the prevalence of HCV in a rural population of a sub-region of Romania and to link it to antiviral treatment. This integrated project of testing-diagnosis-treatment performed in this region had as objectives: micro-elimination of HCV among patients living in this region, prevention of advanced HCV liver disease, prevention of HCV transmission among the healthy population and updating the epidemiological data regarding HCV in this region.

## 2. Materials and Methods

### 2.1. Study Design

We conducted a single-center prospective study based on the strategy of a project designed to educate, test and treat HCV with the aim of eliminating HCV infection in all adults in a village in the Moldavia region in north-eastern Romania—a region considered to have a poor socioeconomic level and with limited access to the healthcare system. The community-based strategy combined an educational campaign, testing and treatment. Our project was carried out from 1 March 2019 to 28 February 2020. To achieve these objectives, it was necessary to include all stakeholders—the local government authorities, associations of health-care providers and patients. A mobile team of gastroenterologists, residents and nurses from the Institute of Gastroenterology and Hepatology, “St Spiridon“ Hospital Iasi was created, undertaking community mobilization with the aid of the local leadership (mayor, village council, general practitioners, teachers and priest). The entire population of the village over the age of 18 was invited to be tested by direct door-to-door communication. In some particular situations, the testing was organized at a household level, in order to perform screening at the level of the whole village. 

### 2.2. Study Assessments

The screening was carried out using rapid diagnostic orientation tests (RDOT) for HCV diagnosis and was performed for all subjects who presented for testing. All patients with HCV-positive antibodies were referred to a tertiary gastroenterology and hepatology department to confirm the active infection, staging and treatment prescription (linkage-to-care). All demographic data, such as age, gender, ethnicity, marital status, employment, education and data on risk factors for HCV infection, were collected through a questionnaire (Table A1 in Appendix A). 

### 2.3. Ethics

This study was approved by the National Ethics Committee, and written informed consent was obtained from each patient in accordance with the principles of the 1975 Declaration of Helsinki.

### 2.4. Statistical Analysis

The collected data were statistically analyzed using the SPSS 20.0 (Chicago, IL, USA) software. The prevalence of HCV-positive subjects was calculated with a 95% confidence interval (CI). Groups were compared using the Chi square test or Fisher’s exact test for categorical variables and by the independent *t* Student test or Mann–Whitney *U* test for continuous variables (depending on data distribution). Most of the investigated variables were calculated using logistic regression and odds ratio (OR) together with the corresponding 95% CI. All statistical tests were two-tailed, with a *p*-value ≤ 0.05 considered statistically significant. 

## 3. Results

### 3.1. Prevalence of HCV Infection

All the village inhabitants—in total, 3507 subjects—were invited to be screened by rapid diagnostic orientation tests. Of these, 2945 (84%) subjects signed the informed consent and were consequently tested and enrolled into the study. The prevalence of positive HCV antibodies in the rural population that presented for testing was 2.64% (Figure 1). The study population consisted of 1973 females and 972 males. The mean age of subjects was 56.1 ± 14.5 years. Detailed baseline demographic characteristics of the study cohort are presented in Table 1. 

Overall, the distribution of subjects according to gender was similar for both negative and positive HCV Ab statuses. Regarding age groups, the prevalence of HCV Ab was found to be higher in those between 80–97 years (30.8% versus 14.1%). Furthermore, the prevalence of HCV was higher in widowed subjects (85.9% versus 12.3%) and lower in those married or living together as a couple (14.1% versus 87.7%). According to social and educational status, the prevalence of positive HCV Ab was lower in employees (7.7% versus 27.2%) and those with university studies (6.4% versus 12.5%).

The prevalence of HCV infection increased markedly with different age groups (*p* < 0.001) (Figure 2).

### 3.2. Cascade of Care 

All subjects with positive HCV antibodies (78, respectively 2,64%) were scheduled for further complete evaluation in a tertiary gastroenterology and hepatology center nearby in order to be linked to care. In total, 66 (85%) subjects were presented for evaluation, and 55 (83%) subjects had detectable HCV RNA. Of these, 54 (98%) patients completed antiviral treatment and 53 (99%) obtained a sustained virological response (SVR). The cascade of care is shown in Figure 3.

### 3.3. Association between Risk Factors and Chronic HCV Infection

According to the addressed questionnaire, three (3.84%) patients were identified with a history of HBV/HDV, five (6.41%) had a history of HCV infection, eight (10.25%) individuals had undergone abortions before 1990, six (7.69%) had experienced multiple surgeries, four (5.12%) had blood transfusions before 1992, eleven (14.10%) patients had multiple dental interventions, two (2.56%) declared sexual contacts with multiple partners, one (1.28%) was using intravenous drugs, three (3.84%) patients had undergone tattooing/piercing procedures, eight (10.25%) shared personal hygiene objects, and no patients had professional exposure to blood products or hemodialysis (Table 2). 

The main risk factors associated with chronic HCV infection were a family history of HCV (OR = 2.23, 95%CI = 1.37–3.5, *p* < 0.0001), blood transfusions performed before 1992 (OR = 3.21, 95%CI = 2.25–4.52, *p* < 0.0001), abortions conducted before 1990 (OR = 1.35, 95%CI = 1.02–1.9, *p* = 0.023), multiple surgical interventions (OR = 1.32, 95%CI = 1.05–1.72, *p* = 0.038) and sharing personal hygiene objects (OR = 1.45, 95%CI = 1.12–1.73, *p* = 0.002) (Table 2). 

## 4. Discussion

Hepatitis C virus infection represents a global public health problem and is a major cause of morbidity and mortality in HCV-infected patients compared to those that are cured or uninfected [18]. It places a large burden on local health systems and economic sectors because it is one of the leading causes of cirrhosis, hepatocellular carcinoma and liver transplantations worldwide [19]. In a short period of about 25 years since its discovery, advances in viral replication and the pathogenesis of viral infection C have made it possible to develop safe therapeutic regimens with 95–100% effectiveness in eradicating the infection, paving the way for a global strategy to eliminate HCV infection [20].

The introduction in 2013 of direct-acting antivirals for the treatment of HCV infection and the achievement of increased rates of sustained virologic response after treatment prompted the World Health Organization to adopt the first ambitious strategy for the Global Health Sector on Viral Hepatitis in 2016, with its main objective being the elimination of HCV infection as a public health threat by 2030 [14,21]. The strategy laid out national targets that include the following objectives: decreasing the incidence of the viral hepatitis by 90%, the diagnosis of 90% of people living with hepatitis C, access to therapy to 80% of diagnosed and eligible patients and a 65% reduction of liver mortality through the integrated actions of awareness, testing and access to treatment [22,23].

The global elimination of HCV has become the ultimate endeavor and final objective since the introduction of DAAs. However, the simple availability of these drugs, which can reduce the burden of HCV infection, is not enough to achieve a real impact on morbidity and mortality, much less to target viral eradication [24]. To this end, WHO has urged countries to develop a national plan for viral hepatitis C by raising public and medical awareness, implementing mass screening to identify infected people, expanding access to treatment to cure all viral patients and developing surveillance programs after viral eradication for people with advanced liver disease, which are the key elements in the so-called “HCV cascade of care” [25]. Therefore, a once fatal disease has become an infection that can be cured with minimal effort, provided that one has access to care and treatment.

A more pragmatic approach is the concept of “micro-elimination”, which involves the elimination of hepatitis C in defined segments of the risk population, as well as in geographical areas (regions, cities, villages) as a strategy to incrementally achieve national elimination [26]. Thus, we consider this work as a small step forward in our public health achievements for the elimination of viral hepatitis by 2030 in Romania. The success of this ambitious goal, at least in countries such as Romania, is possible only by dividing it into different micro-elimination campaigns such as the project we have carried out at a sub-regional level in a population with difficult access to the healthcare system. Wealthier countries with a very low prevalence of HCV infection, such as the Netherlands or Belgium, have also successfully tried to develop a plan to achieve viral micro-elimination [27,28,29].

Although care costs for patients with chronic HCV infection are significant for European health systems, eliminating HCV is not only an urgent financial matter but a matter of human rights, as living with HCV greatly impacts the quality of family life [30]. For the patient, a cure is not only possibly life-saving but also life-changing.

In this screening micro-elimination program conducted in a sub-region of Romania, the prevalence of HCV was lower (2.64%) than previously reported. As far as we know, there have been no similar projects in Europe carried out in rural areas. Globally, there are several HCV micro-elimination projects conducted at the community level. For instance, an Egyptian project in which HCV micro-elimination was also initiated in rural areas had very successful outcomes. During an educational and prevention campaign performed in 73 villages, patients were freely tested, linked to care and received free treatment. Of the 200,000 tested individuals, 34,000 tested positive for HCV antibodies and 14,500 were treated. Of those, 99.9% completed the antiviral treatment and 97% achieved an SVR [31]. In our study, 98% received antiviral therapy, 100% completed the treatment and 99% had an SVR.

For example, a successful HCV micro-elimination campaign was performed in Iceland, a country with a low prevalence of HCV before the advent of direct-acting antivirals. Since 2016, with the help of a testing and treatment campaign, almost all of the 1000 people infected with HCV in the country were diagnosed and linked to treatment. So far, the island is on track to eliminate HCV by this year [32].

Considering other European countries, Scotland has carried out large-scale campaigns to raise awareness of the impact of HCV, which has led to a consistent reduction in the incidence and prevalence of HCV in this area. Scotland uses an integrated combination of pathways to target all groups infected with HCV, especially focusing on the treatment of PWIDs, to prove the concept of “treatment as prevention”. They are on track to achieve micro-elimination by decentralizing healthcare to simultaneously increase accessibility and decrease stigmatization [33].

In Spain, with an HCV prevalence of only 0.22%, the data resulting from a national plan initiated in 2016 showed that HCV testing was performed on 90% of the general population, while testing and linkage to care at-risk groups remained suboptimal [34]. The satisfactory results were associated with a large number of treatments and treatment responses in Spain, in a large project carried out in the “El Dueso” prison between May 2016 and July 2017, where 847 prisoners agreed to be tested. Of these, 110 showed HCV-positive antibodies and 86 were HCV RNA-positive. Only 69 patients were treated, achieving a 96.9% SVR rate [35].

Another example of a well-conducted HCV micro-elimination campaign was in Slovenia, where by following up with patients with congenital bleeding disorders who screened positive for HCV in the 1990s, 98% of the cohort were successfully cured following the micro-elimination approach [36].

At the same time, the prevalence of HCV infection increased markedly with different age groups, but with significant differences for those over 60 years of age. This prevalence picture confirms that viral C infection in Romania is a response to the living conditions before 1989: a deplorable public health situation, the absence of disposable syringes and a special demographic policy that led to a large number of illegal abortions with some tragic consequences.

Additionally, widowed subjects, retired persons or housewives, individuals with a lower level of education and the unemployed in this rural area were more susceptible to chronic HCV infection. These results are consistent with data reported in similar studies conducted in Romania [7], Italy and Spain [37,38,39].

The main risk factors identified in this area for the increased prevalence of HCV are family members known to be positive for HCV and sharing personal hygiene items. At the same time, subjects with abortions conducted before 1990, blood transfusions conducted before 1992, multiple surgical interventions and who shared personal hygiene items show the effects of the inappropriate use of medical or surgical practices on the population.

The data from this study show that a real challenge for people with positive HCV antibodies in this rural area was the access to a tertiary gastroenterology/hepatology center located at a distance for further evaluations. Patients with a positive rapid test who did not show for evaluation were elderly (more than 80 years), affected by comorbidities and considered the distance from their village to our hospital as excessive. Currently, antiviral treatment can only be prescribed by a gastroenterologist or an infectious disease specialist, which are available only in a few large towns. Access to treatment would improve if, in future, DAA therapy could also be prescribed by other healthcare professionals and made available in local pharmacies.

Almost all patients (98%) with detectable viremia were eligible for therapy (all types of treatment, interferon-based regimens), with an SVR rate of 99% for those treated. 

Difficult access to medical care is also an issue for populations with remote healthcare systems. Transportation costs, a low socio-economic level, advanced age, uninsured status and poor collaboration between patients with several comorbidities and physicians for the evaluation of co-medication are real obstacles to evaluating and treating patients that showed HCV-positive antibodies after screening. Due to these situations in disadvantaged areas, our next screening project, which will take place from 2020 to 2024, will target high-risk populations. Barriers in access to treatment must be removed primarily by physicians, who must be aware of the hepatic and extrahepatic consequences of this systemic disease, as well as the high efficacy and safety of treatment and therefore the possibility of HCV elimination [40].

The elimination of hepatitis C worldwide is becoming a possibility, with higher chances of success if micro-elimination strategies based on mass screening are implemented. At the same time, a sustained effort on the part of all stakeholders is required, including governmental authorities at the national and local levels, associations of health-care providers, patients and representatives of at-risk populations. Before attempting nationwide elimination, breaking down national elimination goals into smaller, achievable goals for individual population segments may be more realistic [41].

## 5. Conclusions

This micro-elimination project carried out in a rural area is the first in Romania and among a small number of such projects in the world. Micro-elimination involves fewer resources than large-scale country-level initiatives to eliminate HCV and can represent a boost for small victories that inspire large and ambitious efforts. The development of screening programs is crucial for the accessibility of treatment and the achievement of WHO objectives.

## Figures and Tables

**Figure 1 healthcare-09-00651-f001:**
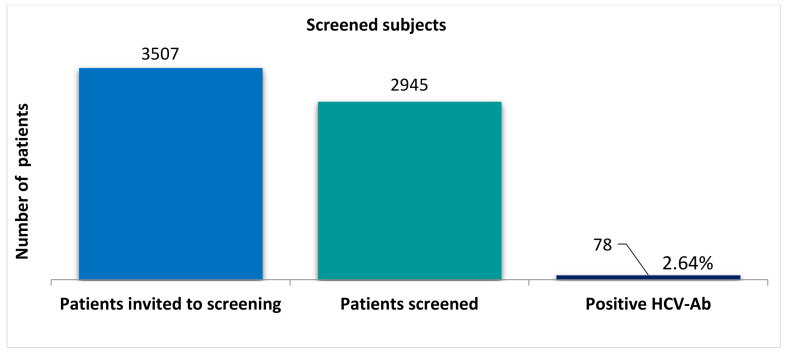
Screened participants for HCV infection, with number of people invited for screening, the total number of screened people and the prevalence of patients with positive HCV antibodies. HCV, hepatitis C virus.

**Figure 2 healthcare-09-00651-f002:**
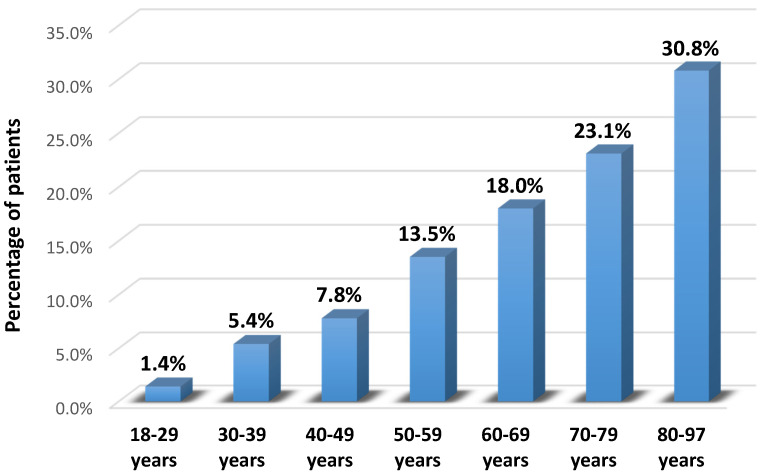
The prevalence of HCV infection in different age groups.

**Figure 3 healthcare-09-00651-f003:**
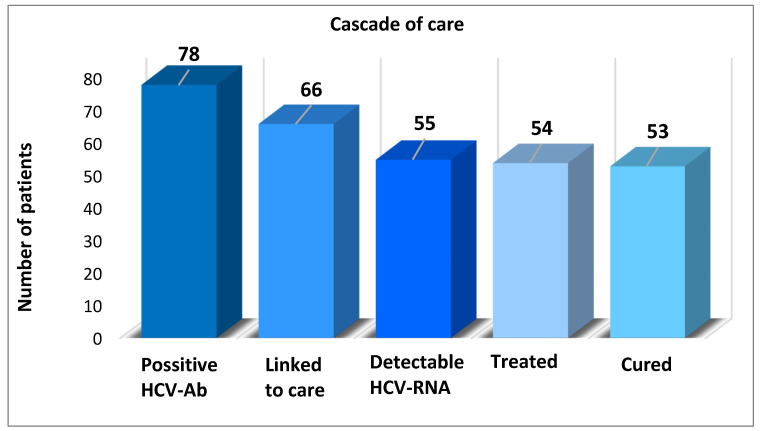
Cascade of care composed of different steps. Positive HCV Ab: Number of people estimated to have viremic HCV infection. Linked to care: Number of patients evaluated for treatment. Detectable HCV-RNA: Number of patients who received a diagnosis of viremic HCV infection. This number excludes patients who were cured of their infection or who had experienced the spontaneous clearance of their infection before 2019. Treated: Number of patients who initiated HCV treatment (all types of treatment, interferon-based regimens). Cured: Number of patients who obtained a sustained virologic response (SVR).

**Table 1 healthcare-09-00651-t001:** Demographic characteristics of the study cohort.

Variable	Patients with Negative HCV Ab (*n* = 2867)	Patients with Positive HCV Ab (*n* = 78)	*p*-Value
Sex, *n* (%)			
Male	947 (33)	25 (32.1)	0.815
Female	1920 (67)	53 (67.9)	
Age, years, *n* (%)			
18–29	284 (9.9)	1 (1.3)	0.163
30–39	211 (7.4)	4 (5.1)	
40–49	358 (12.5)	6 (7.7)	
50–59	552 (19.2)	11 (14.1)	
60–69	614 (21.4)	14 (17.9)	
70–79	443 (15.5)	18 (23.1)	
80–97	405 (14.1)	24 (30.8)	
Marital status, *n* (%)			
Widowed or divorced	353 (12.3)	67 (85.9)	0.037
Married or living together as a couple	2514 (87.7)	11 (14.1)	
Social status, *n* (%)			
Retired persons	1277 (44.5)	44 (56.4)	0.26
Housewives	810 (28.3)	28 (35.9)	
Employees	780 (27.2)	6 (7.7)	
Educational status, *n* (%)			
Subjects without university studies	2510 (87.5)	73 (93.6)	0.188
Subjects with university studies	357 (12.5)	5 (6.4)	

**Table 2 healthcare-09-00651-t002:** Risk factors associated with chronic HCV infection.

Risk Factors	HCV Negative (*N* = 2867) *n* (%)	HCV Positive (*N* = 78) *n* (%)	OR	95% CI	*p* Value
Known HBV/HDV	46 (1.60)	3 (3.84)	0.62	0.3–1.31	0.302
Known HCV (+) family members	47 (1.63)	5 (6.41)	2.23	1.37–3.5	0.0001
Professional exposure to blood products	78 (2.72)	0 (0.00)	0.25	0.11–0.53	0.0001
Abortion before 1990	141 (4.91)	8 (10.25)	1.35	1.02–1.9	0.023
Multiple surgeries	83 (2.89)	6 (7.69)	1.32	1.05–1.72	0.038
Blood transfusions before 1992	45 (1.56)	4 (5.12)	3.21	2.25–4.52	0.0001
Multiple dental interventions	67 (2.33)	11 (14.10)	1.12	0.67–1.45	0.303
Hemodialysis	33 (1.15)	0 (0.00)	0.34	0.06–1.03	0.062
Sexual contacts with multiple partners	133 (4.63)	2 (2.56)	0.88	0.52–1.38	0.615
Intravenous drugs	78 (2.72)	1 (1.28)	0.71	0.4–1.44	0.302
Tattooing/piercing	81 (2.82)	3 (3.84)	1.25	0.72–1.84	0.251
Sharing personal hygiene objects	103 (3.59)	8 (10.25)	1.45	1.12–1.73	0.002

OR, odds ratio; CI, confidence interval; HBV, hepatitis B virus; HCV, hepatitis C virus; HDV, hepatitis D virus.

## Data Availability

The data presented in this study are available on request from the corresponding author. The data are not publicly available because they are the property of the Institute of Gastroenterology and Hepatology, Iasi, Romania.

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
