# Peer review of "Hepatitis C Virus Prevalence and Risk Factors in a Village in Northeastern Romania—A Population-Based Screening—The First Step to Viral Micro-Elimination"

_healthcare, 2021, doi:10.3390/healthcare9060651_

Round 1

Reviewer 1 Report

This manuscript by Huiban L et.al was conducted to evaluate the prevalence of HCV infection and risk factors in a Romanian village population. In this current form of the research article, the following major issues identified

  1. The figure no.2 is not clearly presented. The presentation of figure 2 is very confusing (adding up all these percentage also does not give 100%). I would like to request to please provide the demographic of all the 78 HCV detected participants, such as how many of them are between age 18-29, 30-39 and so on., How many of them are male and female, how many of them are married, single or window, divorced etc.), employment, education, and data on risk factors for HCV infection) and please perform the calculation among them. Form that it will be easily visible for example which age group is more prevalent. Therefore, please provide full demographic profile of 78 HCV detected participants.
  2. Line no 159- 165 is not required and does not make any sense. It explains how the calculation is done in cascade of care. Therefore, please remove line 159-165.
  3. Chapter 3.3., please provide the number of participants having (a) history of HCV, (b) professional exposure to blood products (c) blood transfusions performed (d), abortions undergone (e) multiple surgical interventions and (f) sharing personal hygiene objects and then please provide all these calculation from line 167-173.

Minor comments

  1. There is numerous typo error in the manuscript. Please consider correcting these issues.
  2. Please correct the spelling of percentage in figure 2.

In its current form, the study fails to provide any impact due to its presentation.

Author Response

Response to Reviewer 1 Comments

Point 1: The figure no.2 is not clearly presented. The presentation of figure 2 is very confusing (adding up all these percentage also does not give 100%). I would like to request to please provide the demographic of all the 78 HCV detected participants, such as how many of them are between age 18-29, 30-39 and so on., How many of them are male and female, how many of them are married, single or window, divorced etc.), employment, education, and data on risk factors for HCV infection) and please perform the calculation among them. Form that it will be easily visible for example which age group is more prevalent. Therefore, please provide full demographic profile of 78 HCV detected participants.

Response 1: We thank you for your constructive criticism. As suggested, we corrected figure no. 2 so that it is clear and easy to understand. At the same time, we provide full demographic of all the 78 HCV infection detected participants. Please see Figure No. 2 (page 5, lines 149-151) and Table 1 (page 4, lines 136-138).

Point 2: Line no 159- 165 is not required and does not make any sense. It explains how the calculation is done in cascade of care. Therefore, please remove line 159-165.

Response 2: Thank you for this suggestion. We have removed line 159-165. Please see lines 175-181, page 6.

Point 3: Chapter 3.3., please provide the number of participants having (a) history of HCV, (b) professional exposure to blood products (c) blood transfusions performed (d), abortions undergone (e) multiple surgical interventions and (f) sharing personal hygiene objects and then please provide all these calculation from line 167-173.

Response 3: Thanks for raising a very important aspect. As suggested, we made changes to Table 2 and added the number of participants with risk factors. Please see Table 2, lines 198-200, page 7. We provide all these calculation from line 183-197, pages 6-7.

Minor comments

Point 1: There is numerous typo error in the manuscript. Please consider correcting these issues.

Response 1: The language in the manuscript has been polished.

Point 2: Please correct the spelling of percentage in figure 2.

Response 2: Thank you for this observation. I corrected the spelling of percentage. Please see Figure No. 2 (page 5, lines 149-151).

Reviewer 2 Report

In this study by Huiban et al. “Hepatitis C virus prevalence and risk factors in a village from 2 Northeastern Romania – a population-based screening - the 3 first step to viral micro-elimination” the authors have tried to screen eligible patients for HCV antibody in Romania and those positive have been linked to care for further treatment. It is an interesting study and I commend the authors on the same. I have read with great interest this article but have some concerns in the way the results were presented-

Major concerns-

  1. As the authors will note of the 3507 patients screened only 78 patients tested positive for HCV antibody. Since only very patients tested positive, doing further subset analysis base on marital status, social status, level of education increases the chances of Type 1 error. Unless the authors noted p value of interaction to be significant I would suggest to remove these results since they do not add much value.
  2. I did not see the questionnaire in the supplementary files or as part of the results. It would be of interest to see the type of data collected.
  3. It would be also a good idea to present the baseline characteristics of the two groups- (HCv ab + and HCV ab -ve) and compare and discuss the significant difference in terms of IVDU, educational status etc.
  4. The authors have presented risk factors for HCV infection, but have not provided raw numbers, since only 78 patients tested +, I doubt that these results add value.

Minor-

Perhaps the authors could discuss if any small or major similar projects have been conducted in Europe and their success rates.

I think it is an interesting study and provides great input. However I suggest that the authors relook at their data and conduct more analysis and try to present the results in a more systematic way which will add great value to the study

Author Response

Response to Reviewer 2 Comments

Major comments

Point 1: As the authors will note of the 3507 patients screened only 78 patients tested positive for HCV antibody. Since only very patients tested positive, doing further subset analysis base on marital status, social status, level of education increases the chances of Type 1 error. Unless the authors noted p value of interaction to be significant I would suggest to remove these results since they do not add much value.

Response 1: We have removed the subset analysis as suggested, and perfomed calculations based on marital status, social status and level of education within all screened subjects. Please see lines 152-159, pages 5-6.

Point 2: I did not see the questionnaire in the supplementary files or as part of the results. It would be of interest to see the type of data collected.

Response 2: We added the adressed questionnaire in the supplementary files as suggested. Please see Appendix A, Table A1.

Point 3: It would be also a good idea to present the baseline characteristics of the two groups- (HCV ab + and HCV ab -ve) and compare and discuss the significant difference in terms of IVDU, educational status etc.

Response 3: Thank you for this important suggestion. We added the baseline characteristics of the two groups- (HCV ab + and HCV ab -ve) and discussed the differences between them. Please see Table 1, page 4, lines 139-145, pages 4-5.

Point 4: The authors have presented risk factors for HCV infection, but have not provided raw numbers, since only 78 patients tested +, I doubt that these results add value.

Response 4: As suggested, we provided the raw numbers for risk factors in Table 2, page 7, lines 198-200.

Minor comments

Perhaps the authors could discuss if any small or major similar projects have been conducted in Europe and their success rates.

Response: As far as we know, there are no similar projects conducted in rural population in Europe. We added some valuable studies conducted in other areas than Europe, as well as European projects involving HCV micro-elimination in urban regions. Please see the "Discussion" section, lines 245-277, pages 8-9 and “References” section, lines 415-434, page 11.

Round 2

Reviewer 1 Report

The authors have attended my criticism in a satisfactory manner.

Reviewer 2 Report

The authors have answered all the queries